# Stochastic Expectation Maximization Algorithm for Linear Mixed-Effects Model with Interactions in the Presence of Incomplete Data

**DOI:** 10.3390/e25030473

**Published:** 2023-03-08

**Authors:** Alandra Zakkour, Cyril Perret, Yousri Slaoui

**Affiliations:** 1Laboratoire de Mathématiques et Applications, Université de Poitiers, 11 Boulevard Marie et Pierre Curie, 86962 Futuroscope Chasseneuil, CEDEX 9, 86073 Poitiers, France; 2CeRCA-CNRS UMR 7295, Université de Poitiers, 5 rue T. Lefebvre, MSHS, CEDEX 9, 86073 Poitiers, France

**Keywords:** linear mixed-effects model, interactions, missing data, censored data, EM algorithm, SEM algorithm

## Abstract

The purpose of this paper is to propose a new algorithm based on stochastic expectation maximization (SEM) to deal with the problem of unobserved values when multiple interactions in a linear mixed-effects model (LMEM) are present. We test the effectiveness of the proposed algorithm with the stochastic approximation expectation maximization (SAEM) and Monte Carlo Markov chain (MCMC) algorithms. This comparison is implemented to highlight the importance of including the maximum effects that can affect the model. The applications are made on both simulated psychological and real data. The findings demonstrate that our proposed SEM algorithm is highly preferable to the other competitor algorithms.

## 1. Introduction

The time between the presentation of a stimulus and a participant’s motor response is the oldest and most widely used measure for exploring the functioning of the human mind. In 1869, ref. [1] theorized this duration, called reaction time (RT), as involving three sets of activities: perceptual mechanisms, cognitive processing and motor preparation. Based on the assumption that the first and last sets of processing can be considered as having virtually identical duration for the same task, any change in RTs between two experimental conditions is then interpreted as indicating a change in the duration of cognitive processing. RT is then considered by psychologists as a tool to explore cognitive processing mechanisms ([2]).

Psycholinguistics research on the cognitive mechanisms involved in language recognition or production frequently uses RT as a measure of behavior. Like all scientific disciplines, psycholinguistics relies on hypothesis testing to support its theoretical propositions. Since the early 2000s, researchers have taken up linear mixed-effects models (LMEMs). As described by [3], a LMEM allows for the proper consideration of one of the characteristics of psycholinguistic experiments: the presence of two random-effect variables. Indeed, the experimental structure of the experiments conducted in this field involves having a group of participants process a set of stimuli (see hereafter). Thus, the statistical analysis must allow the inclusion of these two random-effect variables, i.e., participants and items, in the structure of the model. The introduction of LMEM was thus an important methodological advance for psycholinguistics.

There is, however, one point that has not received much attention from psycholinguists. Experimentation with human beings is often subject to many vagaries. Imagine researchers whose goal is to understand how an adult retrieves the spelling of a word from their memory ([4]). They ask a group of thirty participants to write down the names of 150 images presented on a computer screen. A device allows them to measure the time between the appearance of the image on the screen and the first writing gesture of the participant. These researchers can potentially collect 4500 RT values. However, between trials lost for technical reasons, spelling errors, and certain habits during data analysis (e.g., censoring of data greater than two standard deviations from the mean, right-censoring data), a non-negligible number of data are removed. For example, ref. [4] reported removing just over 20% of the data.

The issue of missing and censored data has received relatively little attention in psycholinguistics (see, however, [5,6]). This is especially true since the introduction of LMEMs. Psycholinguists assume that these models can be run on a sample of data with “holes”. The analysis strategy is then of the “keep the case empty” type, ignoring the bias that this introduces in the estimation of the parameters of the model and thus of the decisions taken. The objective of this work is thus to develop algorithms to manage the presence of missing and censored data in these psycholinguistic experiments. Two points are to be taken into account. On one hand, experimental designs in scientific psychology may involve assumptions about interactions between two or more fixed-effect variables. On the other hand, researchers suggest, for theoretical reasons, that all interactions between fixed-effect variables and random-effect variables made possible by the model should be included ([7]). The potential presence of these two types of interactions (fixed–fixed variables; random–fixed variables) are constraints on the development of the missing data procedure.

Let us first recall that the LMEM ([8]) is an extension of the simple linear model that allows both fixed and random effects to be represented. In 1861, the LMEM was introduced under the name of a one-way random-effects model ([9]), that is, a model with one random variable and without any fixed variable. From 1990 and onward, ref. [9] underlined that LMEMs became popular in many research applications including economics, sociology, insurance, agronomy, epidemiology, genetics, etc. They are used in longitudinal data analysis, multilevel modeling and small estimations. The analysis of this type of models is presented in [10]; for more literature about the methodology, theoretical results and software, see the books [11,12,13,14]. LMEMs are fitted and analyzed in R by using the package lme4 or lmertest ([15]).

In [16], the author presented the fixed interactions between two factors in an LMEM, using the maximum product interaction F-test. Ref. [17] was interested in the sample size of an LMEM. He proposed a formula to estimate the sample size based on testing a mixed model that contained fixed interactions. In 2019, ref. [18] proposed an estimation method to recover the principal effects and interactions, because the existing method did not allow the integration of these effects in a mixed data frame. They approved that the proposed method gave optimal results to their applications’ conditions. On the same side, ref. [19] presented the estimation of the fixed interactions. These interactions are normally introduced by the product of the variables, but the algebraic transformations revealed that this technique did not produce a within-unit estimator. A Monte Carlo method confirmed that the FE (fixed-effects) estimator of (x,z) was biased, if one of these variables was correlated with another one. In order to present the interaction between *x* and *z*, it is possible to use the current syntax x∗z. This consideration is called “double-demeaned”; it is less efficient than the standard FE and only works with T<2. For the application in R, the Markov chain Monte Carlo method is applied using the package mcmc. Ref. [20] introduced the MCMCpack package that contains functions to perform Bayesian inference using a posterior simulation for a number of statistical models (https://CRAN.R-project.org/package=MCMCpack (accessed on 14 Jun 2011)), while [21] presented the package MCMCglmm (https://cran.r-project.org/web/packages/MCMCglmm/index.html (accessed on 2 February 2010)) for the MCMC method to fit the generalized linear mixed-models for multiple response vectors. This method was also used by [22] to identify unknown parameters in the biological field, such as detecting the concentration of target molecules, because of the importance of this method for extracting the information. The MCMC method was applied in the work of [23] to solve mechanical problems by using a Bayesian method.

Ref. [24] presented five types of LMEMs by introducing two random variables (see Section A.1), where the models were compared to show the importance of including the maximum effects that could affect the model.

One of the solutions that handles the missing data can be the expectation–maximization (EM) algorithm. It is a very useful algorithm for the estimation of the maximum likelihood function. This method can be a solution when the only data available do not allow the estimation of the parameters ([25]), and/or the expression of the likelihood is analytically impossible to maximize.

In other words, it aims to provide an estimator when the problem comes from the presence of missing data. When the data are incomplete, the knowledge of these values would make it possible to estimate the parameters. The EM algorithm takes its name from the fact that at each iteration, it operates two distinct steps:**Step E:** This step is called the expectation (E) step, where we are interested in finding the expected value of the unobserved or unknown variables given the observed data and the value of the parameters.**Step M:** This step is called the maximization (M) step; in this step, we maximize the expected log-likelihood by using the estimation of the unknown data carried out in the previous step. These parameter estimates are then used to determine the distribution of the unknown variables in the next iteration.

At some points, the expectation or the maximization steps are impossible to apply directly ([26]); from there, the use of an extension form of EM is useful, such as MCEM, where the (E) step is replaced by a Monte Carlo simulation, or SAEM, where the (E) step is replaced by a stochastic approximation. SAEM was a solution of nonlinear mixed-effects models (NLMEM). Ref. [27] proposed a new methodology for maximum likelihood estimation in mixtures of nonlinear mixed-effects models. The resulting MSAEM (mixture SAEM) algorithm is now implemented in the Monolix software tool.

The aim of this paper was to perform the SEM algorithm, under an LMEM by including two types of incomplete data (the censored and the MAR types) and by taking into consideration for the first time the interactions, where our proposed model contains the interactions between fixed variables and fixed–random variables. This document is organized as follows: In Section 2, we present three algorithms based on the expectation–maximization (EM) method, the first one is called the SAEM algorithm, the second is our proposed SEM algorithm and the third one is based on the MCMC method. We also present the incomplete data types, divided into missing data and censored one. In Section 3, we define the proposed model with some specific cases. In Section 4, we introduce a method to achieve the convergence. In Section 5, we compare the results obtained from simulated psychological and real data. In Section 6, we conclude the proposed study with some perspectives and future directions.

## 2. Methodology

### 2.1. EM Algorithm

As previously said, the EM algorithm is a widely used algorithm in the case of incomplete data; in this situation, the maximum likelihood function is difficult or impossible to use to estimate the parameter vector θ of the considered model. We formalize directly an iteration from which we can understand clearly how this algorithm works:−Let $y=(y_{1},\cdots ,y_{n})$ be the independent and identically distributed (i.i.d.) observations of likelihood p(y|θ).−The maximization of logp(y|θ) is impossible.−We consider hidden data $z=(z_{1},\cdots ,z_{n})$ which make the maximization of the likelihood of the complete data possible when known.−As we do not know these data *z*, we estimate the likelihood of the complete data by taking into account all the known information so the estimator is given as follows (E step):
(1)$$\begin{array}{c} Q\left(\theta |\theta_{k-1}\right)=E\left[\log p\left(y,z;\theta |y;\theta_{k-1}\right)\right], \end{array}$$where $\theta_{k-1}$ is the vector of the parameters at iteration k−1.−Finally, we maximize this estimated likelihood to determine the new value of the parameter (M step). Thus, the transition from iteration k−1 to iteration *k* in the algorithm consists in determining the parameters vector at iteration *k*, $\theta_{k}$:
$$\theta_{k}=\arg\max_{\theta }Q\left(\theta |\theta_{k-1}\right).$$
where $\theta_{0}$ is chosen arbitrarily. When one of these two steps are impossible to complete, we can consider an extension form of the EM algorithm such as the SAEM, SEM or MCEM algorithms. In the next subsection, we derive the SAEM algorithm into the formula of the EM algorithm.

#### 2.1.1. SAEM Algorithm

The stochastic approximation expectation maximization (SAEM) algorithm was proposed by [26], in which the E step was replaced by a stochastic approximation. The stochastic approximation algorithm was first introduced by [28] and also used by [29], where the algorithm generates iterates of the form:$$\theta_{k}=\theta_{k-1}-\gamma_{k}\left[h\left(\theta_{k-1}\right)+\epsilon_{k}\right],$$
where $\gamma_{k}$ is a sequence of positive step sizes, *h* is a function of θ, and $\epsilon_{k}$ is a constant such that $\epsilon_{k}=E\left[h\left(\theta_{k}\right)\right]$. From this form, the SAEM algorithm is obtained where the E step of the EM algorithm is divided into two steps; consider the iteration *k*:First, we sample a realization $z_{k}$ of the latent variable from the conditional distribution ($p_{\theta_{k-1}}\left(z|y\right)$) of *z* given *y*, using the value of the parameter $\theta_{k-1}$ at iteration k−1.Second, by using the realization $z_{k}$ from the first step, we update the value of $Q_{k}\left(\theta |\theta_{k-1}\right)$ (see, (Equation 1)) through a stochastic approximation procedure.

Then, the algorithm continues as follows:**Initialization step:** Initialize $\theta_{0}$ in a fixed compact set.

Then, for all k≥1, the *k*th iteration consists in three steps:
**Simulation step:** simulate $z_{k}$ from the conditional distribution $p_{\theta_{k-1}}\left(.|y\right)$.**Stochastic approximation step:** compute the quantity
$$Q_{k}\left(\theta \right)=Q_{k-1}\left(\theta \right)+\gamma_{k}\left(\frac{1}{m(k)}\sum_{j=1}^{m(k)}\log f\left(z_{k}\left(j\right);\theta \right)-Q_{k-1}\left(\theta \right)\right),$$where $m_{k}$ is the number of simulations at each iteration.**Maximization step:** update the parameter value according to $\theta_{k}=\arg\max_{\theta }Q_{k}\left(\theta \right)$.

The SAEM algorithm is more efficient compared to the MCEM, where, at each iteration, the simulation of the missing values is repeated and the ones obtained previously are not used. In the SAEM, all the simulated missing values contribute to the expectation step and that is the advantage of this algorithm where the maximization step is cheaper than the simulation one. Next, we consider the SEM algorithm to show how it differs from an SAEM.

#### 2.1.2. SEM Algorithm

The SEM approach is a specific case of the SAEM algorithm. In the SEM algorithm, the step size is set to zero $\gamma_{k}=0$ and the number of simulations by iteration is constant (usually, $m_{k}=1$).

This method is an extension of the EM algorithm and is a solution when the maximum likelihood is impossible to complete. This algorithm was introduced by [30], where between the steps E and M, they used a stochastic step that consisted in generating a complete sample containing latent variables from the conditional density, based on the observed data. Starting with an initial position $\theta_{0}$, an iteration *k* of the SEM algorithm, where, to each $\theta_{k}$, a $\theta_{k+1}$ is associated, is defined as follows:**Step E:** Compute the conditional density $f(y|x;\theta_{k})$;**Step S:** Draw $z_{k}$ from the conditional distribution, then obtain the complete sample $y_{k}=\left(x,z_{k}\right)$;**Step M:** Update the parameters $\theta_{k+1}$ by maximizing the likelihood function based on the complete vector $y_{k}$.

In the next subsection, we consider the MCEM algorithm to show how a Monte Carlo simulation is used in the EM algorithm.

#### 2.1.3. MCEM Algorithm

In 1990, ref. [31] proposed to replace the expectation step to compute $Q(\theta |\theta_{k-1})$ by a Monte Carlo integration, hence the name MCEM algorithm (Monte Carlo EM). At iteration *k*, the E step is replaced by the following procedure:

**Simulation step (S-step):** generate m(k) realizations $z_{k}\left(j\right)$ (with j=1,…,m(k)) of the missing data vector under the distribution function $p(z;\theta_{k})$

**Monte Carlo integration:** compute the approximation of $Q(\theta |\theta_{k-1})$ according to:$$Q_{k}\left(\theta \right)=\frac{1}{m(k)}\sum_{j=1}^{m(k)}\log f\left(z_{k}\left(j\right);\theta \right).$$The **maximization step** remains unchanged.

As said before, these algorithms can be used in such a way as to handle the unobserved values of the data that are defined in this next subsection.

### 2.2. Incomplete Data

We can talk about incomplete data when the values in our response vector are not all observed for many reasons; we can also say it is a question without an answer.

These types of data are a common problem recognized by statisticians. Ref. [32] presented many solutions to handle this problem based on simple and multiple imputations. Ref. [33] proposed to handle the incomplete data in a hierarchical model by using the SEM algorithm.

In this work, we are interested in data that contain two types of incomplete values: censored data denoted by $Y_{c}$ and missed data $Y_{miss}$. In the next subsection, the censored data are considered.

#### 2.2.1. Censoring

Censored data are any data for which we do not know the exact event time.

There are two types of censored data: right-censored and left-censored.

Right-censored data: these are data where the event has not yet been achieved when the study is finished.

Left-censored data: these are data when the event is achieved before the study starts.

We consider in this paper both censoring types of (left- and right-censoring) data with four percentages (0%, 5%, 10% and 20%). Let *t* be the censoring level and *T* the set of censored indices that can be written as $T=\{\left(i_{1},\ldots ,i_{p+k})\in I,Y_{i_{1},\ldots ,i_{p+k},j}<t\;\mathrm{or}\;Y_{i_{1},\ldots ,i_{p+k},j}>t\right\}$, and let $Y=(Y_{o},Y_{c})$, with $Y_{o}=\left\{Y_{i_{1},\ldots ,i_{p+k},j},\left(i_{1},\ldots ,i_{p+k}\right)\notin T\right\}$ the vector of observed response variables and $Y_{c}=\left\{Y_{i_{1},\ldots ,i_{p+k},j},\left(i_{1},\ldots ,i_{p+k}\right)\in T\right\}$ defining the vector of censored response variables. The second type is the missing data, defined in the next subsection.

#### 2.2.2. Missing Data

In general, missing values are produced when a value in the data is not represented for a given variable, for many reasons that can be linked to the objective of the study (for example, participants do not answer the questions). This appears in many research studies, particularly when collecting data and where participants are studied over a period of time.

At first, studies were developed assuming no missing values. In the late 1980s, with the advancement in technology, that problem attracted the attention of many researchers who wished to study several techniques for handling it. Depending on the reasons for their absence, these values could be divided into three types: missing completely at random (MCAR), missing at random (MAR) and missing not at random (MNAR).

In Section A.2, we define the types of missing value and the methods to generate these types in a simulation study.

In this paper, we are interested in imputing the MAR type of missing data with four percentages (0%, 5%, 10% and 20%), which are crossed with four other percentages of censoring, resulting in 16 cases of incomplete data. For the missing data, we denote by $Y_{o}$ the vector of observed response variable and $Y_{miss}$ the missed one, so $Y=(Y_{o},Y_{miss})$. In the next section, we present the main results of this work.

## 3. Main Results

### 3.1. The Proposed Model

Motivated by psychological data, in which we typically have two random variables, participants designed by *S* and items designed by *I*, we consider a model that contains *p* covariates $\left\{x_{1};x_{2};\ldots ;x_{p}\right\}$ and two random variables. The predicted variable, also named the variable of interest, is denoted by *Y*. Then, we consider the following linear mixed model:(2)$$Y=\beta_{0,s,i}+\beta_{1,s,i}x_{1}+\beta_{2,s,i}x_{2}+\ldots +\beta_{p,s,i}x_{p}+r_{j,i,s},$$with *r* the residual of the model that follows a normal distribution. We give:$$\left\{\begin{array}{cc} \; & \beta_{0,s,i}=\delta_{0}+S_{0,s}+I_{0,i}\\ \; & \beta_{1,s,i}=\delta_{1}+S_{1,s}+I_{1,i}\\ \; & \beta_{2,s,i}=\delta_{2}+S_{2,s}+I_{2,i}\\ \; & \vdots \\ \; & \beta_{p,s,i}=\delta_{p}+S_{p,s}+I_{p,i}. \end{array}\right.$$By a replacement of these considerations in Equation (Equation 2), our model can be rewritten as follows: $$\begin{array}{cc} Y & =\delta_{0}+\delta_{1}x_{1}+\delta_{2}x_{2}+\ldots +\delta_{p}x_{p}\\ \; & +S_{0,s}+S_{1,s}x_{1}+S_{2,s}x_{2}+\ldots +S_{p,s}x_{p}\\ \; & +I_{0,i}+I_{1,i}x_{1}+I_{2,i}x_{2}+\ldots +I_{p,i}x_{p}\\ \; & +r. \end{array}$$By taking
$$X=\left[\begin{array}{ccccc} 1 & x_{1} & x_{2} & \ldots & x_{p+k} \end{array}\right]=\left[\begin{array}{cc} 1 & x_{j} \end{array}\right],\;\delta ={\left(\begin{array}{cc} \delta_{0} & \delta_{j} \end{array}\right)}^{T}$$
and denoting by *T* the transpose matrix,
$$Z=\left[\begin{array}{c} z_{j} \end{array}\right]=\left[\begin{array}{c} x_{j} \end{array}\right],\;u=\left[\begin{array}{c} u_{j} \end{array}\right]={\left[\begin{array}{c} S_{j,l,s}+I_{j,l,i} \end{array}\right]}^{T};$$
$$\mathrm{and}\;e_{jl}=\left[\begin{array}{c} e_{j} \end{array}\right].$$This leads us to rewrite our model as follows:$$Y=X\delta +\sum_{k=1}^{K}z_{k}u_{k}+e=X\delta +Zu+e,$$
where E(Y)=Xδ and $\mathrm{Var}\left(Y\right)=V=ZGZ^{T}+R$, with u∼N(0,G), e∼N(0,R) and $\mathrm{Cov}(u,e^{T})=0$.

Now, by including all the interactions, the general model can be written as:$$Y=F+I_{\mathrm{FF}}+R+I_{\mathrm{FR}}+e,$$
where we take:$$\begin{array}{cc} F & =\underset{\mathrm{fixed}\;\mathrm{variables}}{\underbrace{\delta_{0}+\delta_{1}x_{1}+\ldots +\delta_{p}x_{p}}},\\ I_{\mathrm{FF}} & =\underset{\mathrm{interaction}\;\mathrm{between}\;\mathrm{fixed}-\mathrm{fixed}\;\mathrm{variables}}{\underbrace{\delta_{p+1}x_{p+1}+\ldots +\delta_{p+k}x_{p+k}}},\\ R & =\underset{\mathrm{random}\;\mathrm{variables}}{\underbrace{\left(S_{1,s}x_{1}+I_{1,i}x_{1}\right)+\ldots +\left(S_{p,s}x_{p}+I_{p,i}x_{p}\right)}},\\ I_{\mathrm{FR}} & =\underset{\mathrm{interaction}\;\mathrm{between}\;\mathrm{fixed}-\mathrm{random}\;\mathrm{variables}}{\underbrace{\left(S_{p+1,s}x_{p+1}+I_{p+1,i}x_{p+1}\right)+\ldots +\left(S_{p+k,s}x_{p+k}+I_{p+k,i}x_{p+k}\right)}},\\ \mathrm{and}\;e & =\underset{\mathrm{residuals}}{\underbrace{(S_{0,s}+I_{0,i})+r}}. \end{array}$$We set *k*, the number of interactions, as $k=2^{p}-p-1$ and let j,l∈[1,p+k].

Our random participants variable is given as follows:$$\begin{array}{c} S_{j,l,s} \end{array}∼N\left(0,\Sigma_{S}\right),$$
and the variance–covariance matrix of the participants is written as
$$\begin{array}{c} \Sigma_{S}=\left\{\begin{array}{cc} \sigma_{j,S}^{2} & ;\;\mathrm{if}\;j=l\\ \rho_{S}\sigma_{j,S}\sigma_{l,S} & ;\;\mathrm{otherwise}. \end{array}\right. \end{array}$$We consider that the items random variable also follows a normal distribution:$$\begin{array}{c} I_{j,l,I} \end{array}∼N\left(0,\Sigma_{I}\right);$$
then, the variance–covariance matrix of the items is given by:$$\begin{array}{c} \Sigma_{I}=\left\{\begin{array}{cc} \sigma_{j,I}^{2} & ;\;\mathrm{if}\;j=l\\ \rho_{I}\sigma_{j,I}\sigma_{l,I} & ;\;\mathrm{otherwise}. \end{array}\right. \end{array}$$
For j,l∈[1,p+k], the variance–covariance matrix *G* is equal to:$$\begin{array}{c} G=\Sigma_{u}=\left\{\begin{array}{cc} \sigma_{j,S}^{2}+\sigma_{j,I}^{2} & ;\;\mathrm{if}\;j=l\\ \rho_{S}\sigma_{j,S}\sigma_{l,S}+\rho_{I}\sigma_{j,I}\sigma_{l,I} & ;\;\mathrm{otherwise}. \end{array}\right. \end{array}$$
The goal is to predict our model by determining the parameters δ and *u*; we propose to use Henderson’s linear system ([34]) based on the maximum likelihood function. This approach is presented in Section A.3. The matrix *R* has the form:$$\begin{array}{c} R=\left\{\begin{array}{cc} I & ;\;\mathrm{if}\;j=l\leq n-p-k\\ \sigma_{e_{j}}^{2} & ;\;\mathrm{if}\;j=l>n-p-k\\ 0 & ;\;\mathrm{otherwise}, \end{array}\right. \end{array}$$
where *n* represents the total number of observations. Therefore, with $Z=(z_{1},\ldots ,z_{p+k})$, the variance of *Y* is equal to: $$\begin{array}{ccc} V & = & \left[\left(\sum_{j=1}^{p+k}z_{j}\Sigma_{u_{1},u_{j}}\right)z_{1}+\left(\sum_{j=1}^{p+k}z_{j}\Sigma_{u_{2},u_{j}}\right)z_{2}+\ldots +\left(\sum_{j=1}^{p+k}z_{j}\Sigma_{u_{p},u_{j}}\right)z_{p}+\ldots \right.\\ & & \left.+\left(\sum_{j=1}^{p+k}z_{j}\right)\Sigma_{u_{p+k},u_{j}}z_{p+k}\right]+R. \end{array}$$
The linear system equations of Henderson applied in this study is also presented in Section A.4. In the next subsection, we present some specific cases.

### 3.2. Specific Cases

In this section, we show how the model can be simplified if we take the following two cases: first, only the fixed–fixed interaction part is presented and second, there is no interaction.

#### 3.2.1. Case 1: Fixed–Fixed Interaction

In this case, we consider the interactions between the fixed variables. Then, in the random part, the *u* vector is equal to 0:$$\begin{array}{cc} u & ={\left[\begin{array}{ccc} u_{1} & u_{2}\ldots u_{p} & u_{p+1}\ldots u_{p+k} \end{array}\right]}^{T}\\ \; & ={\left[\begin{array}{ccccccc} 0 & 0 & \ldots & 0 & 0 & \ldots & 0 \end{array}\right]}^{T}; \end{array}$$
therefore, the V matrix is equal to:$$\begin{array}{c} V=R=\left\{\begin{array}{cc} I & ;\;\mathrm{if}\;j=l\leq n-p-k\\ \sigma_{e_{j}}^{2} & ;\;\mathrm{if}\;j=l>n-p-k\\ 0 & ;\;\mathrm{otherwise}. \end{array}\right. \end{array}$$
For more mathematical development, see Section A.4.

#### 3.2.2. Case 2: No Interactions

In this case, there are no interactions between the variables; our model is simplified, where the random part with the fixed–fixed interaction is ignored:$$X=\left[\begin{array}{cc} 1 & x_{j} \end{array}\right];\;\delta ={\left(\begin{array}{cc} \delta_{0} & \delta_{j} \end{array}\right)}^{T};\;Z=\left[\begin{array}{c} z_{j} \end{array}\right];\;u=\left[\begin{array}{c} u_{j} \end{array}\right]={\left[\begin{array}{cccc} 0 & 0 & \ldots & 0 \end{array}\right]}^{T};\;e_{jl}=\left[\begin{array}{c} e_{j} \end{array}\right].$$
The V matrix is reduced to:$$\begin{array}{c} V=R=\left\{\begin{array}{cc} I & ;\;\mathrm{if}\;j=l\leq n-p\\ \sigma_{e_{j}}^{2} & ;\;\mathrm{if}\;j=l>n-p\\ 0 & ;\;\mathrm{otherwise}. \end{array}\right. \end{array}$$
See the simplified system in Section A.4. Next, we present how the SEM algorithm can be applied in this work.

### 3.3. SEM Algorithm

In this paper, we propose to handle missing data and censored problems in the presence of interactions by using the SEM algorithm, proposed by [33]. The stochastic expectation maximization (SEM) algorithm is a method used to estimate the parameters when it is complicated to use the EM algorithm; it is a particular case of multiple imputations (MI).

To understand MI ([35]), we need to know the idea behind a simple imputation (SI) in which the nonobserved data {$Y_{no}$} (missed or censored) are replaced by a value and then the parameters are estimated using known methods to maximize the likelihood function. To obtain a robust estimate, the simple imputation can be repeated several times with different values of {$Y_{no}$} and the results can be combined. This method is called multiple imputations.

We applied the algorithm after crossing the MAR values with the censored ones (see Algorithm 1). In the SEM algorithm ([30]), the values of {$Y_{no}$} are drawn from the conditional distribution of the nonobserved data given the observed ones using the current values of the parameters. We generated samples from $P_{\Theta }\left(Y_{no}|Y_{o}\right)$ where this distribution was calculated using Gibbs’s sampling; the procedure is underlined in Algorithm 1 (the SEM method using Gibbs’s sampling was developed in [33]). For more information about the expectation maximization algorithm, see [36]. In the next subsection, we present the extension form of the SEM that leads us to the SAEM algorithm.
**Algorithm 1** SEM algorithm: *N* is the number of iterations of the SEM algorithm, *M* is the burn-in level, $Y_{1},\ldots ,Y_{n}$ is the response vector, *G* is the number of iterations of Gibbs sampling, $Y[u_{i}]$ is the response vector with respect to the *i*th random variable, and fixed is the summation of the fixed effects with the fixed interaction part.**Input:** *N*, *M*, $Y_{1},\ldots ,Y_{n}$, and *G*.
  1:  Random initialization of Θ
  2:  **for** j=1,…,N **do**
  3:      **for** g=1,…,G **do**
  4:          draw $e^{\left(g\right)}$ from $N(0,\sigma_{e}^{2}$, $\mathrm{lower}=\min\left(e\right)-\mathrm{fixed}-u^{\left(g-1\right)}$, $\mathrm{upper}=\max\left(e\right)-\mathrm{fixed}-u^{\left(g-1\right)}$)
  5:          draw $u_{1}^{\left(g\right)}$ from $N(0,\Sigma_{u_{1}}$, $\mathrm{lower}=\max(\min\left(Y\left[u_{1}\right]\right)-u_{2}^{\left(g-1\right)}-e^{\left(g\right)})$, $\mathrm{upper}=\min(\max\left(Y\left[u_{1}\right]\right)-u_{2}^{\left(g-1\right)}-e^{\left(g\right)}))$
  6:          draw $u_{2}^{\left(g\right)}$ from $N(0,\Sigma_{u_{2}}$, $\mathrm{lower}=\max(\min\left(Y\left[u_{2}\right]\right)-u_{1}^{\left(g\right)}-e^{\left(g\right)})$, $\mathrm{upper}=\min(\max\left(Y\left[u_{2}\right]\right)-u_{1}^{\left(g\right)}-e^{\left(g\right)}))$
  7:      **end for**
**output:**     
$Y_{no}$ obtained from sampled ($u^{\left(G\right)}$,$e^{\left(G\right)}$)
  8:      $\Theta_{j+1}={\mathrm{argmax}}_{\Theta }L\left(\Theta |Y_{o},Y_{no}\right)$
  9:  **end for**
10: 
$\hat{\Theta }={\left(N-M\right)}^{-1}\sum_{j=M+1}^{N}\Theta_{j}$
**output:**  
$\hat{\Theta }$


### 3.4. SAEM Algorithm

In the SEM algorithm, $\Theta_{j+1}$ (the parameter at state j+1) depends only on $\Theta_{j}$ and $Y_{j}$. By taking *T* the operator of the EM algorithm and *M*, the operator which associates $\Theta_{j+1}$ with $Y_{j}$ via step M, the updated parameter $\Theta_{j+1}$ can be written as follows (see [30]):$$\Theta_{j+1}=T\left(\Theta_{j}\right)+V\left(\Theta_{j},Y_{j}\right),$$
where $V\left(\Theta_{j},Y_{j}\right)=M\left(Y_{j}\right)-T\left(\Theta_{j}\right)$.

The SAEM algorithm (see also Algorithm 2) is an extension form of the SEM algorithm. Starting with an initial position $\Theta_{0}$, the iteration of the SAEM algorithm, to which for each $\Theta_{j}$, a $\Theta_{j+1}$ is associated, is defined by the equation:$$\Theta_{j+1}=T\left(\Theta_{j}\right)+\gamma_{j}V\left(\Theta_{j},Y_{j}\right),$$
where $\gamma_{j}$ is the step size, which is a decreasing sequence of positive real numbers starting with $\gamma_{0}=1$, $\gamma_{j}\mapsto 0$ when j↦∞. The theoretical and practical performance values of the SAEM algorithm are strongly dependent on the speed of convergence towards 0 and the regular decreasing of the sequence $\gamma_{j}$, which explains the importance of choosing the step size.

Ref. [37] performed many tests to see which sequence was more efficient, and they obtained two types of sequences that were significant, one of a slow mode:$$\gamma_{j}=\cos\left(\frac{j\times \pi }{2N}\right),$$
the other type was of a linear mode:$$\gamma_{j}=-\frac{\left(j+1\right)}{N}.$$ Ref. [38] chose the decreasing sequence from their previous experiments [39]; they took that sequence as
$$\gamma_{j}=\cos\left(N\times \alpha \right){𝟙}_{\left\{0\leq j\leq 20\right\}}+\frac{c}{\sqrt{j}}{𝟙}_{\left\{j>20\right\}},\;\mathrm{where}\;\cos\left(20\times \alpha \right)=\frac{c}{\sqrt{20}}=0.3.$$That sequence was also chosen by [40,41] in a nonlinear mixed model as 1 at the first iteration and ${\left(j-N\right)}^{-1}$ during the last iterations.
**Algorithm 2** SAEM algorithm: *N* is the number of iterations of the SEM algorithm, *M* is the burn-in level, $Y_{1},\ldots ,Y_{n}$ is the response vector, *G* is the number of iterations of Gibbs’s sampling, $\gamma_{j}$ is the decreasing sequence, $Y[u_{i}]$ is the response vector with respect to the *i*th random variable, and fixed is the summation of the fixed effects with the fixed interaction part.**Input:** *N*, *M*, $Y_{1},\ldots ,Y_{n}$, and *G*.
  1:  Random initialization of Θ
  2:  **for** j=1,…,N **do**
  3:      **for** g=1,…,G **do**
  4:          draw e(g) from N(0,σe2, lower=min(e)−fixed−u(g−1), upper=max(e)−fixed−u(g−1))
  5:          draw u1(g) from N(0,Σu1, lower=max(min(Y[u1])−u2(g−1)−e(g)), upper=min(max(Y[u1])−u2(g−1)−e(g)))
  6:          draw u2(g) from N(0,Σu2, lower=max(min(Y[u2])−u1(g)−e(g)), upper=min(max(Y[u2])−u1(g)−e(g)))
  7:      **end for**
**output:**      $Y_{no}$ obtained from sampled ($u^{\left(G\right)}$,$e^{\left(G\right)}$)
  8:      $\Theta_{j+1}={\mathrm{argmax}}_{\Theta }\gamma_{j}\times L\left(\Theta |Y_{o},Y_{no}\right)$
  9:  **end for**
10:  $\hat{\Theta }={\left(N-M\right)}^{-1}\sum_{j=M+1}^{N}\Theta_{j}$
**output:**  $\hat{\Theta }$


When applying these two algorithms, we were confronted with a problem of convergence caused by the presence of the interaction between the fixed and random variables. Therefore, in the next section, we present how to handle this problem by using the Hamiltonian Monte Carlo algorithm.

## 4. Convergence of Parameters

While introducing the fixed–random effect in the application section, we faced a convergence problem in the SEM and SAEM algorithms when using the standard form of Monte Carlo (MC). In order to solve this problem, we propose to use the hybrid MC algorithm ([42]), which uses the symmetric Metropolis–Hastings algorithm to accept or reject a proposal based on the Hamiltonian function, hence the name of the algorithm, the Hamiltonian Monte Carlo algorithm (HMC). In the next subsection, we present how to implement the HMC algorithm.

### Implementation

The Hamiltonian function H(p,q) is written in term of the joint density π(p,q):H(p,q)=−logπ(p,q),
which decomposes into two terms:$$\begin{array}{cc} H(p,q) & =-\log\pi (p|q)-\log\pi (q)\\ \; & =K(p,q)+V(p,q). \end{array}$$The first term corresponds to the density of the target distribution, the second one is determined by the target distribution when K(p,q) is unconstrained and must be specified by the implementation.

The acceptance probability of moving from state $\left(p_{0},q_{0}\right)$ to $\left(p_{i},q_{i}\right)$ is determined using the Metropolis–Hastings algorithm ([43,44]), with $Q(p_{i},q_{i}|p_{0},q_{0})$ being the probability density defining each proposal:$$a\left(p_{i},q_{i}|p_{0},q_{0}\right)=\min\left(1,\frac{Q\left(p_{0},q_{0}|p_{i},q_{i}\right)\pi \left(p_{i},q_{i}\right)}{Q\left(p_{i},q_{i}|p_{0},q_{0}\right)\pi \left(p_{0},q_{0}\right)}\right).$$Referring to the symmetric Metropolis ([45]), the acceptance probability is simplified to the form:$$\begin{array}{cc} a(p_{i},q_{i}|p_{0},q_{0}) & =\min\left(1,\frac{\pi (p_{i},q_{i})}{\pi (p_{0},q_{0})}\right)\\ \; & =\min\left(1,\frac{\exp(-H\left(p_{i},q_{i}\right))}{\exp(-H\left(\left(p_{0},q_{0}\right)\right)}\right)\\ \; & =\min(1,\exp\left(-H\left(p_{i},q_{i}\right)+H\left(p_{0},q_{0}\right)\right)). \end{array}$$In this study, we considered that we had evidence of the convergence towards stationarity due to the Markov Chain (see, [45,46]).

In the next section, we present some numerical applied results based on a psychological simulation and then real data.

## 5. Numerical Experiment

This section aims to compare the SEM algorithm proposed in this paper with other methods in the presence of incomplete data in a complete LMEM. The method was first applied to simulated data, then to real ones.

### 5.1. Simulated Data

The simulation data were created from an experimentally obtained database. In order to explore the cognitive process of retrieval in memory of the spelling of French words, 30 participants had to handwrite the label of 150 drawings of objects, constituting a database of 4500 RTs ([4]). After removing the RTs corresponding to technical errors, spelling errors, and censoring data greater than 2.5 standard deviations from the mean, 3434 values remained. Ref. [4] performed a linear regression analysis involving nine fixed-effects variables. In the present work, we retained two of them. The first was an estimate of the age at which the image name was acquired in childhood (age of acquisition, AoA hereafter). This factor is one of the most important predictors of picture-naming RTs ([47]. The second fixed-effects variable corresponded to the number of letters in a picture label (Lett_cat). We performed a median split to obtain a categorical variable. Thus, image labels with a number of letters lower than the sample average were considered as short words and those with a higher number of letters were considered as long words. The interaction between these two fixed-effects variables was included in the model. Finally, there were two random effect variables: items, i.e., the 150 image labels produced by the 30 participants, and the participants themselves. The fixed–random interaction between AoA and the participants was also included in the LMEM.

Therefore, the model could be written as follows:$$\begin{array}{cc} \mathrm{RTs} & =b_{0}+b_{1}\times \mathrm{AoA}+b_{2}\times \mathrm{Lett}\text{\_}\mathrm{cat}+b_{3}\times \mathrm{AoA}\times \mathrm{Lett}\text{\_}\mathrm{cat}\\ \; & +\;\mathrm{participants}+\mathrm{items}+\mathrm{AoA}\times \mathrm{participants}+e. \end{array}$$In this section, we illustrate the proposed algorithm by comparing the results to the SAEM and MCMC algorithms using different proportions of incomplete data. The comparison was performed by computing the mean absolute error (MAE), which measured the errors between the reference and the predicted vectors (with Θ as the reference value and $\hat{\Theta }$ as the predicted one); the standard equation of the MAE is given by:$$\mathrm{MAE}=\frac{1}{n}\sum_{i=1}^{n}\left|{\hat{\Theta }}_{i}-\Theta_{i}\right|.$$We also computed the linear correlation between the parameters $\left(\mathrm{LCor}\;=\;\frac{\mathrm{Cov}({\hat{\Theta }}_{i},\Theta_{i})}{\sigma \left({\hat{\Theta }}_{i}\right)\sigma \left(\Theta_{i}\right)}\right)$ and the Spearman rank correlation (RCor = $1-\frac{6\sum d^{2}}{n(n^{2}-1)}$, where $d^{2}$ is the square of the difference in the ranks of the two coordinates).

In this numerical section, we provide the 16 cases to see the performance of the methods in the absence of one of these two considered types of unobserved values (MAR type/censored data) or in the presence of a low or high percentage. In addition, we used the simulated data of 4500 observations after testing our proposed methods on a larger number of observations. We obtained the same results as those for 4500 values.

After the implementation of our algorithm, we checked the convergence by illustrating the values of the estimated parameters obtained at each iteration; this is presented in Figure 1.

In Figure 2, we plotted the normalized residuals of each case. We observed a similar dispersion between the standard simulated data and the SEM algorithm, while we noticed a small but visible scattering by implementing the SAEM algorithm, and finally, we observed that when we used the MCMC method, the residuals were dispersed far from the normal line.

Our results are visualized quantitatively in Table 1. Based on the MAE value, we can observe that the SEM gave the lowest values in all cases except for the case of 5%
MAR and 20% censoring, where the SAEM (8.108) beat the SEM (8.118) by a very narrow margin. By observing the full vectors, the SEM gave the best parameter values in two cases: (0%, 10%) and (20%, 5%). In the other cases, we observed that with a high percentage of MAR (20%), b${}_{2}$ and b${}_{3}$ in the SAEM method gave the closest values while for the other parameters, it was the SEM algorithm. In all cases, we see that the MCMC was not a good choice either for the parameters values or for the other measurements. Moreover, we can see that for LCor (respectively, Rcor), the SEM and the SAEM algorithms gave approximately the same results 0.999 (respectively, 1) in all cases. This showed that these two methods had a small difference except for the case of (20%, 5%) where the RCor of the SAEM algorithm decreased from 1 to 0.9285. The MCMC had an RCor ranging between a maximum value of 1 and a minimum value of 0.738; that maximum value was obtained with 5% of MAR and the absence of censored data.

### 5.2. Real Data

In this section, we applied the three algorithms to the database used to create the simulated data. Of the 4500 RTs recorded during the experiment (30 participants producing the names of 150 items), 115 were removed by right censoring (2.56% of MNAR). Ref. [4] also removed 951 values, creating 21.13% of missing MCAR/MAR data. By applying the three proposed algorithms with some missing data to solve the problem in the presence of fixed and fixed–random interactions, we obtained the results presented in Table 2. The results were compared with respect to the initial vector where the missed values were treated by keeping the cases empty (KE). Interestingly, this is how missing and censored data are handled in psycholinguistic studies.

## 6. Conclusions

In this study, we proposed an algorithm based on the stochastic expectation maximization (SEM) to estimate the parameters under a linear mixed-effects model (LMEM) that contained fixed–fixed and fixed–random interactions in the presence of different percentages of incomplete data.

The simulated and real data showed that the proposed SEM algorithm gave the best results compared to other competitors, where the bias of the parameters was smaller than the bias in the SAEM and MCMC algorithms.

The simulation results were obtained by using statistical software R (see Section A.5 for some source code).

We plan to extend this approach by considering a more complicated level of interaction such as random–random effects and the interactions between more than two variables. Another direction could be to consider an extension of the present work by considering a generalized model under the logistic regression by taking one for the NA values and zero for the observed ones.

## Figures and Tables

**Figure 1 entropy-25-00473-f001:**
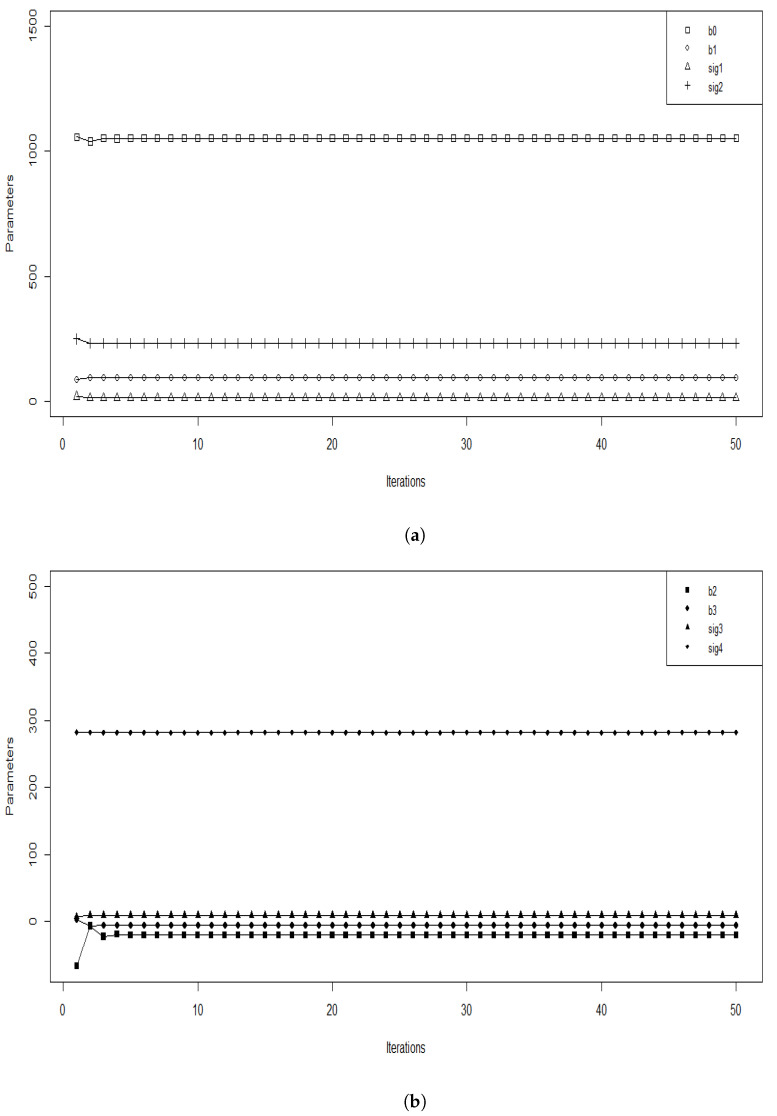
Value of the parameter estimates at each the SEM iteration with a burn-in period M = 10, we separate the parameters according to their values in (**a**), we plot b0, b1, sig1 and sig2 and in (**b**), we plot b2, b3, sig3 and sig4.

**Figure 2 entropy-25-00473-f002:**
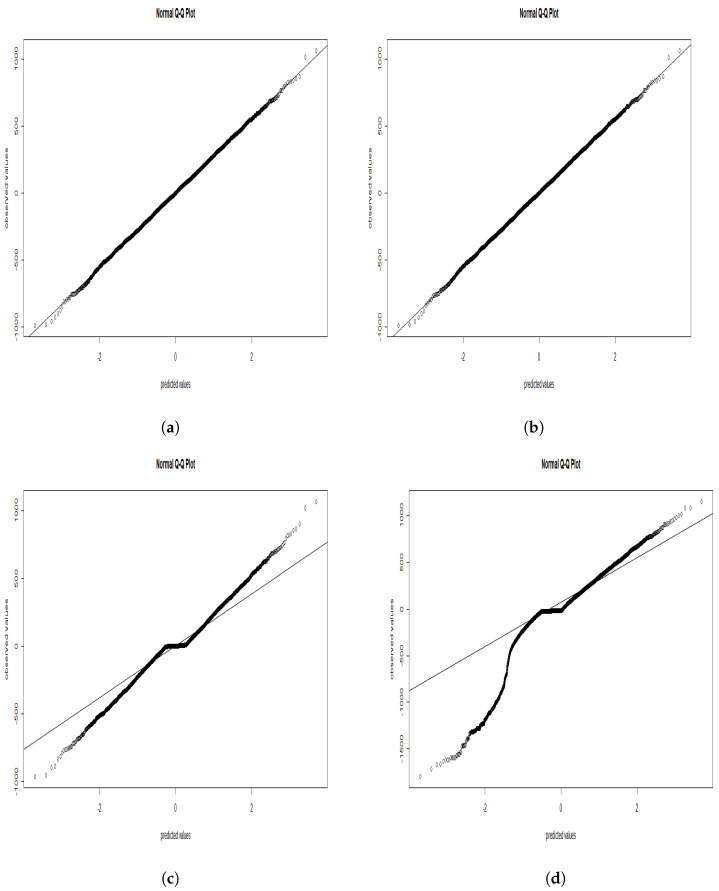
Normalized residual presented; (**a**)-standard simulated data, (**b**)-SEM algorithm, (**c**)-SAEM algorithm, (**d**)-MCMC algorithm.

**Table 1 entropy-25-00473-t001:** Comparison of the SEM algorithm with the SAEM and MCMC algorithms in the complete model with the different percentages of incomplete data where the number of iterations was 50. sig${}_{1}$, sig${}_{2}$, sig${}_{3}$ and sig${}_{4}$ represent the standard deviations of the participants, items, the I${}_{\mathrm{FR}}$ part and the error, respectively. The percentages on the left present the MAR, and on the right are the percentages of censored data.

	(0%,0%)			(0%,5%)			(0%,10%)			(0%,20%)		
${\hat{\Theta }}_{i}$	**Reference**			**SEM**	**SAEM**	**MCMC**	**SEM**	**SAEM**	**MCMC**	**SEM**	**SAEM**	**MCMC**
b${}_{0}$		**1045**		1044	**1045**	1068	**1044**	1043	1010	**1047**	**1047**	822.8
b${}_{1}$		**93.16**		**93.54**	88.51	48.82	**93.91**	89.37	41.25	**93.51**	88.84	77.96
b${}_{2}$		**−52.28**		**−51.30**	−42.33	−36.75	**−52.50**	−47.30	51.49	−30.27	**−51.66**	18.26
b${}_{3}$		**2.80**		**3.17**	−0.18	6.92	**4.43**	5.52	−14.44	**−1.79**	5.02	−18.20
sig${}_{1}$		**14.57**		**14.65**	14.43	5.29	**14.65**	12.54	12.56	**14.70**	12.17	11.30
sig${}_{2}$		**230.8**		**231.7**	225.9	350.5	**231.1**	223.0	430.5	**233.2**	219.0	538.7
sig${}_{3}$		**9.79**		**9.81**	9.84	2.80	**9.80**	8.13	8.61	**9.81**	9.12	8.94
sig${}_{4}$		**282.0**		**282.0**	275.5	283.1	**282.0**	269.4	277.8	**282.0**	253.5	253.9
MAE		**0**		**0.421**	3.637	28.07	**0.505**	4.628	51.84	**3.935**	6.575	83.64
LCor		**1**		**0.999**	**0.999**	0.992	**0.999**	**0.999**	0.971	**0.999**	**0.999**	0.915
RCor		**1**		**1**	**1**	0.904	**1**	**1**	0.738	**1**	**1**	0.833
	(5%,0%)			(5%,5%)			(5%,10%)			(5%,20%)		
${\hat{\Theta }}_{i}$	**SEM**	**SAEM**	**MCMC**	**SEM**	**SAEM**	**MCMC**	**SEM**	**SAEM**	**MCMC**	**SEM**	**SAEM**	**MCMC**
b${}_{0}$	**1044**	1038	1004	1069	**1037**	1019	1069	**1036**	963	1078	**1038**	780
b${}_{1}$	**93.30**	94.33	93.79	87.57	**90.04**	52.82	88.07	**90.90**	45.29	86.52	**90.75**	80.88
b${}_{2}$	−46.69	**−53.57**	−49.05	**−49.96**	−42.94	−28.23	**−52.68**	−47.94	55.58	−32.18	**−50.21**	26.67
b${}_{3}$	**1.04**	4.58	6.55	8.27	**1.27**	8.63	9.70	**6.75**	−11.90	**3.82**	5.61	−15.79
sig${}_{1}$	**15.19**	13.65	16.59	**15.25**	17.02	8.34	15.36	**13.93**	16.19	**15.79**	12.80	4.10
sig${}_{2}$	**231.0**	223.7	213.6	**229.8**	218.7	332.4	**228.9**	216.0	412.7	**232.4**	211.9	518.8
sig${}_{3}$	**9.94**	7.93	11.07	**9.97**	8.63	5.05	**9.95**	7.52	10.61	**10.11**	8.42	6.56
sig${}_{4}$	**282.4**	281.4	347.6	**282.6**	275.0	349.4	**282.6**	268.7	342.2	**282.6**	253.2	317.6
MAE	**1.146**	2.646	16.78	**4.970**	5.516	34.43	**5.089**	6.287	62.03	8.118	**8.108**	88.97
LCor	**0.999**	**0.999**	0.996	**0.999**	**0.999**	0.991	**0.999**	**0.999**	0.971	**0.999**	**0.999**	0.911
RCor	**1**	**1**	**1**	**1**	**1**	0.928	**1**	**1**	0.738	**1**	**1**	0.809
	(10%,0%)			(10%,5%)			(10%,10%)			(10%,20%)		
${\hat{\Theta }}_{i}$	**SEM**	**SAEM**	**MCMC**	**SEM**	**SAEM**	**MCMC**	**SEM**	**SAEM**	**MCMC**	**SEM**	**SAEM**	**MCMC**
b${}_{0}$	1070	**1037**	981	1069	**1035**	999	1070	**1034**	938	1080	**1037**	759
b${}_{1}$	86.59	**94.05**	90.52	87.58	**90.27**	50.902	88.10	**90.80**	46.43	86.29	**90.33**	81.31
b${}_{2}$	−45.96	**−50.70**	−62.01	**−45.00**	−39.50	−44.14	**−47.79**	−44.57	45.21	−28.34	**−48.01**	23.38
b${}_{3}$	6.79	**5.26**	12.25	6.69	**1.46**	14.45	8.13	**6.78**	−8.83	**2.53**	5.81	−16.26
sig${}_{1}$	**14.80**	24.43	4.36	**14.97**	27.35	25.00	**15.15**	22.09	30.51	**15.93**	21.39	18.64
sig${}_{2}$	**228.7**	220.1	205.1	**230.0**	215.3	323.1	**229.1**	212.5	401.1	**232.5**	208.7	506.5
sig${}_{3}$	**9.79**	11.46	5.94	**9.89**	12.10	6.336	**9.88**	10.16	13.87	**10.11**	10.50	8.73
sig${}_{4}$	**282.4**	279.6	398.0	**282.5**	273.2	389.6	**282.5**	267.2	379.8	**282.5**	251.8	354.3
MAE	**5.608**	4.699	30.12	**5.387**	8.279	40.15	**5.424**	8.191	68.85	8.747	9.636	93.20
LCor	**0.999**	**0.999**	0.990	**0.999**	**0.999**	0.987	**0.999**	**0.999**	0.969	**0.999**	**0.999**	0.908
RCor	**1**	**1**	0.904	**1**	**1**	0.976	**1**	**1**	0.833	**1**	**1**	0.833
${\hat{\Theta }}_{i}$	**SEM**	**SAEM**	**MCMC**	**SEM**	**SAEM**	**MCMC**	**SEM**	**SAEM**	**MCMC**	**SEM**	**SAEM**	**MCMC**
b${}_{0}$	**1045**	1035	896	**1045**	1044	908	**1045**	1036	857	**1051**	**1039**	687
b${}_{1}$	**93.34**	94.05	100.8	**93.70**	80.78	64.43	**94.05**	90.14	57.35	**92.78**	89.41	91.59
b${}_{2}$	−35.09	**−38.36**	2.54	**−34.54**	−128.87	21.66	−36.11	**−34.79**	97.19	−19.91	**−39.24**	85.55
b${}_{3}$	−1.82	**1.90**	−16.60	**−1.55**	39.61	−13.36	−0.36	**4.00**	−31.09	−4.73	**3.67**	−42.98
sig${}_{1}$	**12.12**	26.88	29.49	**12.44**	11.70	39.89	**12.62**	25.58	33.25	**13.62**	26.53	22.49
sig${}_{2}$	**230.9**	211.4	204.7	**231.8**	118.2	315.9	**231.3**	204.4	387.9	**233.1**	200.9	485.3
sig${}_{3}$	8.84	11.33	**8.95**	**8.98**	5.15	11.48	**9.03**	10.91	10.93	**9.41**	11.76	6.227
sig${}_{4}$	**282.3**	276.2	465.9	**282.3**	265.3	453.6	**282.3**	264.0	442.2	**282.4**	249.2	411.6
MAE	**3.261**	8.002	57.07	**3.374**	32.83	67.38	**2.994**	10.905	93.01	**6.343**	12.47	117.2
LCor	**0.999**	**0.999**	0.969	**0.999**	0.991	0.969	**0.999**	**0.999**	0.944	**0.999**	0.998	0.873
RCor	**1**	**1**	0.976	**1**	0.9285	0.928	**1**	**1**	0.761	**1**	**1**	0.833

We marked in bold for each case the closest value to the reference among the three considered methods (the reference vector here was the simulated one that did not contain any missed value).

**Table 2 entropy-25-00473-t002:** Comparison of the initial parameters in the KE method versus the parameters estimated after the application of the SEM, SAEM and MCMC algorithms. The initial parameters presented here are the ones obtained from the model that contained 1066 NA (NA represent the incomplete data; here, we had 115 censored data and 951 missing data.

$\hat{\Theta }$	KE	SEM	SAEM	MCMC
b${}_{0}$	852.5	875.7	892.334	1298
b${}_{1}$	146.6	134.2	120.4	−161.4
b${}_{2}$	158.6	126.1	79.63	−335.3
b${}_{3}$	−58.12	−42.08	−26.72	146.6
sig${}_{1}$	184.3	196.0	203.5	216.9
sig${}_{2}$	133.0	127.8	94.87	221.05
sig${}_{3}$	37.05	28.67	32.84	66.13
sig${}_{4}$	279.5	302.5	262.9	516.6

## Data Availability

Data is unavailable due to privacy.

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
