# Peer review of "Stochastic Expectation Maximization Algorithm for Linear Mixed-Effects Model with Interactions in the Presence of Incomplete Data"

_entropy, 2023, doi:10.3390/e25030473_

Round 1

Reviewer 1 Report

The paper proposes a new Stochastic Expectation Maximization (SEM) algorithm for Linear Mixed Effects Model (LMEM) with interactions in presence of incomplete data. The authors present three algorithms based on the well-known Expectation Maximization (EM) method: 1) the Stochastic Approximation Expectation Maximization (SAEM) algorithm, 2) the proposed SEM algorithm, and 3) the Monte Carlo EM (MCEM) algorithm. The paper briefly introduces two main types of incomplete data (missing and censored data), describes the statistical model and discusses the convergence of parameters. The performance of the proposed method is illustrated using both simulated and real data sets. The paper appears to be well-written and comprehensively referenced.

I have the following comments on the manuscript.

Lines 114-115. Explain \theta (is it a vector of parameters?), \theta_k and \theta_{k-1}.

Lines 118-119. What is \epsilon_k?

Line121-122. It should be $z$ and $y$ instead of z and y.

Line 123. What is Q_k? It has not been introduced before.

Lines 130-131. What is m(k)?

Line 167. I would say that the left bracket before i_{p+k} is not needed here.

Lines 192-193. Insert a space before “our model can be rewritten as follows” and remove the space after this phrase. Insert + before \delta_p x_p in “Y=…”. What is r? What do u ~ (0,G) and e ~ (0,R) mean (also, see Appendix)?

Lines 223-224 (see also Algorithm 2). T, N and M are used both in mathematical (italic font) and text modes. It should be consistent.

Figures 1-2. It is hard to read the text on the plots.

Author Response

Dear Editor, Associate Editor and Reviewer,

Thank you for your kind letter of "entropy-2241243-Major Revisions" on the 20th February, 2023. The title of the manuscript is: "Stochastic Expectation Maximization algorithm for Linear Mixed Effects Model with interactions in presence of incomplete data" (by Alandra Zakkour, Cyril Perret and Yousri Slaoui). Based on your  comments, we have made careful modifications to the original manuscript. 

We hope that these revisions improve the paper such that you deem it worthy and appropriate for publication. 

Sincerely yours.

Alandra Zakkour, Cyril Perret and Yousri Slaoui.

Here are our responses to the reviewer comments.

The reviewer's comment : 1. The paper mainly looks like a repacking of the available methods. What is new here?

The authors's answer : We are grateful for your comments. In the revised version of the paper, the following sentence was introduced in the lines 109-111 : "The aim of this paper is to perform the \texttt{SEM} algorithm,\hspace{0.5ex}under a LMEM by including two types of incomplete data (the censored and the MAR type) and by taking into consideration for the first time the interactions, where our proposed model contains the interactions between fixed variables and fixed-random variables".

The reviewer's comment : 2. The reviewer did not understand why Hamiltonian Monte Carlo is used. Why more efficient MCMC techniques, such DREAM, DRAM, EnKF-MCMC etc. are not employed? Since no convergence results are shown, the reviewer can not verify its efficiency.

The authors's answer : We thank you very much for your comments. In the revised manuscript, the following sentence was introduced in the lines 246-247 : "While introducing the fixed-random effect in the application section, we faced a convergence problem with the SEM and SAEM algorithms using the standard form of Monte Carlo (MC)".

The reviewer's comment : 3. Please provide evidence regarding the convergence of the MCMC chain.

The authors's answer : We are very grateful for your suggestions. In the revised version of the paper, the following sentence was introduced in the lines 253-254 : "In this study, we consider that we have the evidence of convergence towards the stationary due to a Markov Chain (See, Andrieu et al. (2003); Roberts and Rosenthal (2004))".

The reviewer's comment : 4. What is the necessity of the appendix? What is new here?

The authors's answer : We thank you very much for your comments. In the revised manuscript, the following sentence was introduced in the line 403 : "In the following section, we present how Matuschek et al. (2017) defined the models by including the maximum effects. It is also interesting to show the types of missing data with the implementation methods. Then, the approach of Henderson is introduced and our model matrices' to estimate the parameters are defined. Finally, we present some source code of the SEM algorithm".

The reviewer's comment : 5. Why did you use the MCMC? What has been inferred? Where are the posterior densities?

The authors's answer : We thank you very much for your comments. MCMC method is used by applying Gibbs sampling in the aim of comparing this method to SEM and SAEM where also the sampling method was Gibbs. In the conclusion we have mentioned that the MCMC method was the worst among the three proposed methods. The posterior densities was not presented here, we defined in the appendix our estimated model matrices' for the linear system of  Henderson from which we can determine the parameters' estimation.

The reviewer's comment : 6. The numerical examples should be justified. Please provide details regarding their importance.

The author's comment : We are grateful for your comments. In the revised manuscript, in the lines 253-256, the following sentence was added : "In this numerical section, we have provided the 16 cases to see the performance of the methods in the absence of one of these 2 considered types of unobserved values (MAR type/censored data) or in the presence of a low or high percentage. Plus, we are using a simulated data of 4500 observations after testing our proposed methods on a largest number of observations. We obtained the same results as the 4500 values".

The reviewer's comment : 7. The English quality of the manuscript should be improved. The reviewer found a couple of grammatical mistakes and typos. Careful proofreading is requested.

The author's comment : We are grateful for your comments. In the revised manuscript, We associated our paper to an English expert.

The reviewer's comment : 8. The figures’ quality and readability should be improved.

The author's comment : We are very grateful for your comments. In the revised manuscript, we have changed the figures' scales.

The reviewer's comment : 9. The authors are requested to introduce the recently related published papers. The following paper should be suggested in the introduction.

• Rational Design of Field-Effect Sensors Using Partial Differential Equations, Bayesian Inversion, and Artificial Neural Networks \url{https://doi.org/10.3390/s22134785}

• Bayesian Inversion with Open-Source Codes for Various One-Dimensional Model Problems in Computational Mechanics \url{https://doi.org/10.1007/s11831-022-09751-6}

The author's comment : We are very grateful for your comments and suggestions. In the revised manuscript, we have added these two references to the Introduction in lines 84-87 in the following sentence : " This method is also used by Khodadadian et al. (2022) to identify the unknown parameters in the biological field like detecting the concentration of target molecules, and this is because of the importance of this method in extracting the information. MCMC method is applied in the work of Noii et al. (2022) to solve mechanical problems by using a bayesian method".

Again, we appreciate all your insightful comments. Thank you for taking the time and energy to help us improve the paper.

Sincerely yours

Author Response

Dear Editor, Associate Editor and Reviewer,

Thank you for your kind letter of "entropy-2241243-Major Revisions" on the 20th February, 2023. The title of the manuscript is: "Stochastic Expectation Maximization algorithm for Linear Mixed Effects Model with interactions in presence of incomplete data" (by Alandra Zakkour, Cyril Perret and Yousri Slaoui). Based on your  comments, we have made careful modifications to the original manuscript. 

We hope that these revisions improve the paper such that you deem it worthy and appropriate for publication. 

Sincerely yours.

Alandra Zakkour, Cyril Perret and Yousri Slaoui.

Here are our responses to the reviewer comments.

The reviewer's comment : 1. Lines 114-115. Explain $\theta$ (is it a vector of parameters?), $\theta_k$ and $\theta_{k-1}$.

The authors's answer : We thank you very much for your comments. In the revised version, we added the following definitions in the lines 120-127 : 

• "the parameters vector $\theta$"  

• "with $\theta_{k-1}$ is the vector of the parameters at iteration  $k-1$."

• "the parameters vector at iteration $k$, $\theta_{k}$."

The reviewer's comment : 2. Lines 118-119. What is $\epsilon_k$?

The authors's answer : We thank you very much for your comments. In the revised version of the paper, the following sentence was introduced in the line 133 : "and $\epsilon_{k}$ is a constant such that  $\epsilon_{k}=\mathbb{E}[h(\theta_{k})]$."

The reviewer's comment : 3. Line 121-122. It should be $z$ and $y$ instead of z and y.

The authors's answer : We apologize for this mistake. We corrected it in the lines 126-127.

The reviewer's comment : 4. Line 23. What is $Q_{k}$? It has not been introduced before.

The authors's answer : We thank you very much for your comments. In the revised version of the paper, we defined this notation in the line 138.

The reviewer's comment : 5. Lines 130-131. What is $m(k)$?\\

The authors's answer :

We apologize for not mentioning this notation. In the revised manuscript, line 146, we have added the following sentence : " with $m_{k}$ is the number of simulations at each iteration."

The reviewer's comment : 6. Line 167. I would say that the left bracket before $i_{p+k}$ is not needed here.

The author's comment : We thank you very much for your comments. We have corrected this mistake in the revised manuscript in the line 183. 

The reviewer's comment : 7. Lines 192-193. Insert a space before “our model can be rewritten as follows” and remove the space after this phrase. Insert + before $\delta_p x_p$ in “$Y=\ldots$”. What is $r$? What do $u ~ (0,G)$ and $e~(0,R)$ mean (also, see Appendix)?

The author's comment : We apologize for the non-detailed information. We have defined each notation in the revised manuscript in the line 207.

The reviewer's comment : 8. Lines 223-224 (see also Algorithm 2). T, N and M are used both in mathematical (italic font) and text modes. It should be consistent.

The reviewer's comment : We apologize for these mistakes. We have corrected the mistakes in the revised manuscript in the line 238.

The reviewer's comment : 9. Figures 1-2. It is hard to read the text on the plots.

The author's comment : We are very grateful for your comments. In the revised manuscript, we have changed the figures' scales for Figure 1 and Figure 2.

Again, we appreciate all your insightful comments. Thank you for taking the time and energy to help us improve the paper.

Sincerely yours

Round 2

Reviewer 2 Report

The reviewer thanks the authors for considering the comments carefully. The paper is recommended for publication.